# Optimization of Fermentation Conditions for Enhanced Single Cell Protein Production by *Rossellomorea marisflavi* NDS and Nutritional Composition Analysis

**DOI:** 10.3390/foods14173066

**Published:** 2025-08-30

**Authors:** Hui Zhang, Wenwen Zhang, Wen Zhang, Minghan Yin, Lefei Jiao, Tinghong Ming, Xiwen Jia, Moussa Gouife, Jiajie Xu, Fei Kong

**Affiliations:** 1Microbial Development and Metabolic Engineering Laboratory, School of Marine Science, Ningbo University, Ningbo 315211, China; zh032220@163.com (H.Z.); 2211130071@nbu.edu.cn (W.Z.); 2411130131@nbu.edu.cn (W.Z.); yinmhan@163.com (M.Y.); jiaolefei@nbu.edu.cn (L.J.); mingtinghong@nbu.edu.cn (T.M.); jiaxiwen@nbu.edu.cn (X.J.); gouife@nbu.edu.cn (M.G.); 2Key Laboratory of Aquacultural Biotechnology, Ministry of Education, Ningbo University, Ningbo 315211, China

**Keywords:** *Rossellomorea marisflavi* NDS, single cell protein, fermentation conditions, nutritional composition

## Abstract

Microbial proteins offer a sustainable alternative for animal nutrition. *Rossellomorea marisflavi* NDS, a bacterium isolated from seawater, was previously identified as a promising candidate due to its high protein content. This study aimed to enhance its single cell protein production through systemic fermentation optimization. Single-factor optimization in shake flask determined the optimal conditions to be: a salinity of 20‰ NaCl, a temperature of 32 °C, and an initial pH of 7.3, and a medium composed of 1% (*w*/*v*) corn flour, 1% peptone, 0.3% beef extract, and 0.2% KCl. Scaling up to a 10 L bioreactor demonstrated that a two-stage agitation strategy (150 rpm for the first 20 h followed by 180 rpm for the remaining 12 h) enhanced single cell protein yield. Furthermore, allowing the pH to fluctuate freely was more beneficial for protein production than maintaining a constant pH of 7.3 ± 0.02. Under these optimized conditions, the biomass composition (wet weight) was determined to be 2.3767 ± 0.0205% crude ash, 15.6013 ± 0.0082% crude protein, 0.1023 ± 0.0026% crude lipid, and 2.6997 ± 0.0021% carbohydrates. Amino acid analysis revealed a rich profile, with lysine and glutamic acid being the predominant essential and non-essential amino acids, respectively. Fatty acids analysis indicated that C14:1n5 was the most dominant. These findings underscore the potential of *R. marisflavi* NDS as a high-quality dietary protein supplement and provide a solid foundation for its industrial-scale production.

## 1. Introduction

Proteins are essential natural biomacromolecules that play crucial roles in biological systems [1,2]. The rapidly growing global demand for protein-rich animal feed has created an urgent need to develop novel dietary protein sources [3]. This challenge is particularly pronounced in aquaculture and livestock production, where conventional protein supplements, such as soybean meal and fishmeal, are increasingly inadequate [4]. Recently, microbial proteins have emerged as a promising alternative, garnering significant research interest [5]. They offer three key advantages: (1) Nutritional superiority, they contain complete amino acid profiles comparable to those of animal proteins, along with essential vitamins and minerals [6]; (2) Production efficiency, rapid microbial growth allows for high yields in compact facilities, regardless of season or geography [7]; and (3) Environmental sustainability, they can be produced using organic waste streams, which reduces production costs while promoting circular economy principles [8,9,10]. This combination of nutritional value, scalable production, and ecological benefits positions microbial proteins as a transformative alternative to conventional protein sources.

Microbial proteins can be synthesized by various microorganisms, including microalgae, bacteria, yeast, and fungi, each offering distinct metabolic and bioprocessing advantages. Although microalgal biomass is known for its high protein content (50–70% dry weight, DW) and favorable amino acid profile, its direct utilization is often limited by poor digestibility [11]. Yeast, which is rich in protein, amino acids, and trace elements, is widely used in aquatic feed and industrial applications [12]. Notable species include *Kluyveromyces marxianus* and *Saccharomyces cerevisiae* [13,14]. However, the thick and rigid cell wall of yeast restricts its broader utilization. Among microbial protein producers, bacteria stand out as a critical platform due to their unique advantages in bioprocessing applications. Key bacterial species utilized in this context include *Escherichia coli* [15], lactic acid bacteria [16], *Methylococcus capsulatus* [17], *Clostridium autoethanogenum* [18], and *Bacillus subtilis* [19].

Microbial proteins provide high-quality protein and essential nutrients with functional benefits, typically containing <5% fat [20]. Their nutritional profiles vary by microbial type: Microalgal proteins demonstrate species-dependent amino acid profiles [21], fungal proteins (30–50% protein) are rich in glutamic acid, lysine, and leucine but limited in methionine and tryptophan, while also providing vitamins, minerals, and 7–10% nucleic acids [22]. Bacterial proteins are particularly noteworthy, reaching up to 80% protein content with excellent leucine, lysine, and valine, often comparable to fishmeal in methionine and lysine [20]. The fermentation process further enhances their nutritional quality by (1) improving protein digestibility, (2) increasing essential amino acid availability, and (3) producing beneficial metabolites including short chain fatty acids (SCFAs) and antioxidants like glutathione [23]. These combined attributes make microbial proteins highly valuable as functional feed additives.

In our previous study, we isolated a yellow-pigmented bacterial strain from a marine environment. Phylogenetic analysis using 16S rDNA sequencing (GenBank Accession No. SUB15557226) identified the isolate as *Rossellomorea marisflavi*, designated as strain NDS, which has been deposited in the China General Microbiological Culture Collection Center (CGMC No. 32287). Notably, this strain demonstrated an exceptionally high protein content of 42.37% (*w*/*w*, DW) [24]. In the present study, we systematically optimized the fermentation process to maximize single cell protein production by *R. marisflavi* NDS. We first determined the optimal cultivation conditions (including salinity of NaCl, temperature, and initial pH) as well as the ideal culture medium composition (carbon sources, nitrogen sources, and inorganic salts) using single-factor optimization at the shaking flask level. Subsequently, we confirmed the effectiveness of these optimizations through large-scale fermentation experiments in a 10 L bioreactor, investigating the effects of agitation speed and pH control on protein accumulation. Additionally, we conducted a nutritional composition analysis (proximates, amino acids, and fatty acids) of the NDS strain to evaluate its nutritional value. These findings provide a solid foundation for advancing the industrial application of *R. marisflavi* NDS, highlighting its potential as a promising candidate for dietary protein supplementation.

## 2. Materials and Methods

### 2.1. Strain and Culture Conditions

*R. marisflavi* NDS was isolated from the central part of Xiangshan Port in Zhejiang Province, at a depth of 0.5 m below the sea surface [24]. The strain was initially inoculated on beef extract-peptone solid medium (containing 0.3% (*w*/*v*) beef extract, 1.0% (*w*/*v*) peptone, 20‰ NaCl, and 1.0% (*w*/*v*) agar) and incubated at 37 °C for 12 h. For seed culture preparation, the activated NDS was inoculated into 30 mL of beef extract peptone liquid medium (BPM) and incubated at 37 °C with shaking at 150 rpm for 12 h. Subsequently, the seed culture was inoculated into 100 mL of BPM at a 1% (*v*/*v*) inoculum size for 22 h to obtain the final fermentation broth. Bacterial density was determined by measuring the optical density at 600 nm (OD_600_) using a Multiskan SkyHigh Spectrophotometer (Thermo Scientific, Singapore). To assess the growth curve of *R. marisflavi* NDS, OD_600_ was measured at 2 h intervals during fermentation.

### 2.2. Determination of Protein Content

After cultivation for 22 h in 250 mL conical flasks containing 100 mL medium, bacterial cells were harvested by centrifugation at 10,000 rpm for 5 min at 4 °C. The cell pellets were washed three times with phosphate-buffered solution (PBS) to remove residual medium, a step which also eliminated negligible amounts of extracellular proteins. Proteins were extracted using a commercial protein extraction kit (Shanghai Beibo Biotechnology Co., Ltd., Shanghai, China), and the concentration was quantified using a BCA protein assay kit (Suzhou Biyuntian Biotechnology Co., Ltd., Suzhou, China) by measuring absorbance at 562 nm. All experiments were performed in triplicate with biological replicates (independent cultures). To assess the protein accumulation profile of *R. marisflavi* NDS during fermentation, protein content was measured at 2 h intervals.

### 2.3. Optimization of Cultivation Condition

Single-factor experiments were conducted to investigate the effects of cultivation conditions on bacterial growth and protein accumulation in NDS. The experiments were performed using a series of 250 mL flasks. Each flask contained 100 mL of BPM media inoculated with 1% (*v*/*v*) of the bacterial seed culture. Three process parameters were evaluated: salinity of NaCl (10, 15, 20, 25, and 30‰), incubation temperature (28, 30, 32, 34, 36, and 37 °C), and initial pH (6.3, 6.8, 7.3, 7.8, and 8.3). The strains were shaken at 150 rpm for 22 h. After fermentation, the OD_600_ and protein content were measured according to the aforementioned method (Section 2.1 and Section 2.2). All experiments were performed in triplicate with biological replicates.

### 2.4. Optimization of Medium Composition

Based on the optimal fermentation conditions determined in Section 2.3, single-factor experiments were conducted using the BPM as a baseline to investigate the effects of medium composition on bacterial growth and protein accumulation in NDS. The specific parameters optimized included: carbon sources (1.0% *w*/*v* corn flour, bran, carboxymethylcellulose sodium (CMC-Na), straw powder, microcrystalline cellulose (MCC), glucose, sucrose, fructose, trehalose, D-mannose, formic acid, acetic acid, methanol, ethanol and mannitol); corn flour concentrations (0.5, 1.0, 1.5, 2.0, 2.5, and 3.0% *w*/*v*); nitrogen sources (1.0% *w*/*v* peptone, yeast, soybean meal, tryptone, soybean, urea (CH_4_N_2_O), ammonium chloride (NH_4_Cl), and various combinations of peptone and yeast (1% peptone + 1% yeast, 1% peptone + 0.5% yeast, 0.5% peptone + 1% yeast, 0.5% peptone + 0.5% yeast)); and inorganic salts and their concentrations (0.20% *w*/*v* MgSO_4_·7H_2_O, KCl, K_2_HPO_4_, MnSO_4_, CaCl_2_·2H_2_O; inorganic salt concentrations: 0.10, 0.15, 0.20, 0.25, and 0.30% *w*/*v*). After 22 h of fermentation, OD_600_ and protein content were measured as described in Section 2.1 and Section 2.2, with all experiments performed in triplicate with biological replicates.

### 2.5. Fermentation Condition Optimization in a 10 L Bioreactor

Scaled-up fermentation was conducted in a 10 L bioreactor (Biotech-10BG, China) with a 6 L working volume using the optimized fermentation conditions from previous sections, where the seed culture was inoculated at 1% (*v*/*v*) under constant aeration (3 L/min). We first established growth and protein accumulation profiles to determine the optimal harvest time, then compared control strategies for pH (uncontrolled fluctuation vs. active maintenance at pH 7.3 ± 0.02 with 1 M hydrochloric acid (HCl)/sodium hydroxide (NaOH) and agitation (constant 150 or 180 rpm vs. a two-phase strategy: 150 rpm for 0–20 h followed by 180 rpm in the late-log phase).

### 2.6. Proximate Analysis

Following fermentation under predetermined optimal conditions in the bioreactor, bacterial cells were harvested for nutritional composition analysis. Moisture content was determined by direct drying at 103 °C until constant weight was achieved. Ash content was measured by calcining samples in a muffle furnace at 550 °C. Crude protein content was quantified using the standard Kjeldahl method [25], involving digestion in sulfuric acid (H_2_SO_4_) with a catalyst, distillation after NaOH addition, absorption of released ammonia (NH_3_) in boric acid with a mixed indicator, and titration with standardized HCl. Total crude lipids were extracted from hydrolyzed samples using petroleum ether, followed by solvent distillation, drying, and gravimetric quantification [26]. Carbohydrate content was determined via anthrone-sulfuric acid method [27], with absorbance measured at 620 nm using glucose as the standard.

### 2.7. Amino Acid Analysis

Amino acid analysis was performed using an L-8900 automatic amino acid analyzer (Hitachi, Tokyo, Japan). Samples were acid hydrolyzed with 6 M HCl at 110 °C for 24 h, followed by separation via ion-exchange chromatography and post-column derivatization with ninhydrin for detection at 570 nm. For sulfur-containing amino acids (methionine and cysteine), samples were oxidized with performic acid prior to hydrolysis, concerting them to methionine sulfone and cysteic acid, respectively, before analogous chromatography separation and derivatization. Tryptophan was analyzed separately by reversed-phase high performance liquid chromatography (HPLC) with UV detection: Samples were first hydrolyzed under alkaline conditions using 4 M lithium hydroxide at 110 °C for 20 h under nitrogen protection, separated on a C18 column (250 × 4.6 mm, 5 μm) with a gradient mobile phase of 0.1% trifluoroacetic acid in water and acetonitrile (flow rate: 1.0 mL/min; temperature: 25 °C), and detected at 280 nm. Quantification for all amino acids was performed using the external standard method.

### 2.8. Fatty Acid Analysis

Fatty acid analysis was conducted in accordance with the national standard method GB 5009.168-2016. In brief, precisely weighed samples (0.1 g) were spiked with 2 mL of triundecanoin (C11:0, TAG) internal standard solution. Total lipids were extracted via hydrochloric acid hydrolysis followed by diethyl ether extraction. The extracts were then subjected to alkaline saponification and methylation to generate fatty acid methyl esters (FAMEs), which were separated and quantified by gas chromatography (GC).

An Agilent 8890 GC system (Agilent Technologies Inc., Santa Clara, CA, USA) equipped with a CD-2560 chromatographic column (100 m × 0.25 mm × 0.20 μm, Anpel, Shanghai, China) was used for separation. High-purity helium was employed as the carrier gas at a constant flow rate of 0.80 mL/min. The injector temperature was set to 270 °C, and a 1 μL sample was injected with a split ratio of 100:1. Following injection, the oven temperature was executed as follows: initial hold at 100 °C for 13 min; then increased to 180 °C at 10 °C/min and hold for 6 min; gradual ramp to 200 °C at 1 °C/min and maintained for 20 min; final increase to 230 °C at 4 °C/min and hold for 20 min.

### 2.9. Statistical Analysis

All data are expressed as mean ± standard deviation (SD) of triplicate biological replicates. Statistical significance was determined using one-way analysis of variance (ANOVA) followed by Tukey’s test for pairwise multiple comparisons, with a *p*  <  0.05 considered significant. Analyses were performed using SPSS 27 (IBM Inc., Armonk, NY, USA).

## 3. Results

### 3.1. Temporal Profiles of Bacterial Growth and Protein Accumulation During Fermentation

As illustrated in Figure 1, *R. marisflavi* NDS exhibited biphasic growth kinetics, characterized by an initial growth phase (0–14 h) with increasing OD_600_ values that peaked at 14 h, followed by a subsequent decline phase. These distinct growth phases suggest active proliferation during the initial 14 h period, followed by culture maturation. Notably, the protein accumulation profile displayed asynchronous dynamics relative to biomass growth. The peak in protein synthesis (approximately 22–24 h) occurred substantially later than the logarithmic growth phase, suggesting that protein synthesis occurs during a post-exponential phase, potentially associated with the production of secondary metabolites in this strain. In the present study, a cultivation period of 22 h was identified as the optimal harvest time to maximize protein recovery for downstream processing in shake flask cultures.

### 3.2. Effects of Cultivation Condition on Protein Accumulation and Bacterial Growth

#### 3.2.1. Effect of NaCl Salinity

As shown in Figure 2A, bacterial growth and protein biosynthesis exhibit a pronounced NaCl-dependent response. Specifically, *R. marisflavi* NDS cultured at a salinity of 20‰ NaCl achieved maximum biomass accumulation (OD_600_) and protein production. Both lower and higher NaCl concentrations inhibited cellular growth and protein synthesis, with bacterial proliferation being particularly affected. Thus, a salinity of 20‰ NaCl was determined as the optimal concentration.

#### 3.2.2. Effect of Temperature

The effects of different temperatures, ranging from 28 to 37 °C, on the growth and protein production of *R. marisflavi* NDS are illustrated in Figure 2B. The highest protein production (1007.60 ± 9.32 μg/mL) and bacterial growth were recorded at 32 °C. These values were significantly higher than those observed at other temperatures.

#### 3.2.3. Effect of Initial pH

In the present study, we systematically evaluated the effects of initial pH (6.3, 6.8, 7.3, 7.8, and 8.3) on the growth characteristics and protein production in *R. marisflavi* NDS. As demonstrated in Figure 2C, strain NDS exhibited pH-dependent growth characteristics with maximal biomass accumulation observed at pH 6.8, followed by pH 7.3, while significant growth inhibition occurred at pH 6.3 (*p* < 0.05). Notably, the pattern of protein accumulation differed from the growth trend, with maximal accumulation (1026.05 ± 7.73 μg/mL) observed at pH 7.3. Although protein content at pH 6.8 remained significantly higher than at extreme pH levels (6.3 and 8.3) (*p* < 0.05), it was still lower than that under the optimal production condition. Consequently, the initial pH was set at 7.3 for subsequent experiments.

### 3.3. Effects of Medium Composition on Protein Accumulation and Bacterial Growth

#### 3.3.1. Effect of Carbon Source

In this study, we systematically evaluated the effects of fifteen carbon sources, each at a concentration of 1.0% (*w*/*v* or *v*/*v*), on the growth characteristics and protein accumulation of strain NDS, using BPM as the control check (CK). As shown in Figure 3A and Appendix A, corn flour exhibited the most remarkable cultivation performance, yielding 1135.54 ± 20.19 μg/mL of protein within 22 h of cultivation. This represents a significant 12.40% increase compared to the CK group (*p* < 0.05). Notably, all tested monosaccharides, disaccharides, alcohols as well as organic acids, displayed strong growth inhibition, with suppression rates exceeding 50% (Appendix A).

Recognizing that an inappropriate carbon source concentration could led to suboptimal C/N ratios, we conducted a gradient experiment using trehalose as a representative carbon source (tested concentrations: 0, 0.001, 0.005, 0.01, 0.05, 0.10, 1, 3, and 5%). Appendix A demonstrates growth suppression across all trehalose concentrations tested, with significant inhibition persisting even at minimal doses (*p* < 0.05). Owing to severe growth inhibition, the bacterial biomass yield was substantially reduced, resulting in nearly undetectable protein accumulation after 22 h of cultivation. Consequently, only OD_600_ values could be obtained for certain carbon source supplementation groups. Although other polysaccharides exhibited less growth inhibition, their protein yields remained significantly lower than those in the CK group (*p* < 0.05).

Further optimization revealed a concentration-dependent effect of corn flour (tested at 0.5, 1.0, 1.5, 2.0, 2.5, and 3.0% (*w*/*v*)) on growth and protein production (Figure 3B). Maximum productivity was achieved with 1.0% supplementation, where both OD_600_ and protein content were significantly higher than those at other concentrations (*p* < 0.05).

#### 3.3.2. Effect of Nitrogen Source

We initially investigated the potential of replacing peptone in the BPM with alternative organic and inorganic nitrogen sources, including soybean meal, tryptone, soybean flour, CH_4_N_2_O, and NH_4_Cl. All substitutes demonstrated significant growth inhibition throughout the cultivation period (Appendix A).

Given that peptone and yeast are conventional complex nitrogen sources for microbial culture, we subsequently evaluated various combinations of peptone and yeast extract at different concentrations (1.0% peptone, 1.0% yeast, 1.0% peptone + 1.0% yeast, 1.0% peptone + 0.5% yeast extract, 0.5% peptone + 1.0% yeast, and 0.5% peptone + 0.5% yeast). However, none of the tested formulations showed satisfactory performance, with both OD_600_ and protein accumulation being severely inhibited (65–75% reduction versus CK, *p* < 0.05) (Figure 3C).

The results indicated that 1.0% (*w*/*v*) peptone as the sole nitrogen source supported optimal growth and achieved the highest protein production (1138.91 ± 26.79 μg/mL). In contrast, yeast extract alone demonstrated an inhibitory effect on protein biosynthesis (295.15 ± 14.39 μg/mL). These findings conclusively establish 1.0% peptone as the optimal nitrogen source for this strain.

#### 3.3.3. Effect of Inorganic Ions

Following the optimization of carbon and nitrogen sources, we investigated the effects of inorganic salts (0.2% *w*/*v* MgSO_4_·7H_2_O, KCl, K_2_HPO_4_, MnSO_4_, and CaCl_2_·2H_2_O) compared to CK group on the growth and protein production of *R. marisflavi* NDS. As shown in Figure 3D, a significant positive correlation was observed between protein accumulation and growth (OD_600_). Specifically, supplementation with either KCl or MgSO_4_·7H_2_O resulted in significantly higher biomass production compared to other treatments (*p* < 0.05). The KCl treatment group exhibited the highest protein yield (1330.89 ± 30.74 μg/mL) among all experimental conditions, followed by K_2_HPO_4_. In contrast, the addition of CaCl_2_·2H_2_O markedly inhibited both cell growth (OD_600_ < 0.1) and protein accumulation (349.47 ± 28.77 μg/mL). These results demonstrate that KCl not only effectively promotes the growth of *R. marisflavi* but also significantly enhances its protein biosynthesis efficiency.

Through optimization of KCl concentration, our results demonstrated that while the addition of 0.25% and 0.30% KCl achieved maximal biomass accumulation, these concentrations yielded significantly lower protein production compared to the 0.20% supplementation group (*p* < 0.05). Notably, cultures supplemented with 0.20% KCl achieved peak protein content (1326.79 ± 3.90 μg/mL), representing a 30–60% increase over other concentrations (*p* < 0.05), despite not reaching the highest cell density (Figure 3E). A comprehensive evaluation of both growth performance and protein productivity established 0.20% (*w*/*v*) KCl as the optimal supplementation level.

### 3.4. Effects of Fermentation Condition in a 10 L Bioreactor on Protein Accumulation and Bacterial Growth

#### 3.4.1. Temporal Profiles of Bacterial Growth and Protein Accumulation During Fermentation in a 10 L Bioreactor

Based on the results of the aforementioned single-factor cultivation experiments conducted in shake flasks, this study established the optimal culture conditions for *R. marisflavi* NDS. The optimal conditions were context-dependent and based on the tested conditions: a salinity of 20‰ NaCl, a constant temperature of 32 °C, and an initial pH of 7.3. The optimized medium composition was determined to be: 0.3% (*w*/*v*) beef extract, 1.0% corn flour, 1.0% peptone, and 0.2% KCl. Using these fermentation conditions, a 6 L culture medium was prepared, and a comparative fermentation experiment was conducted in a 10 L bioreactor to evaluate the growth and protein accumulation of *R. marisflavi* NDS in a scale-up fermentation system. As shown in Figure 4, the results indicated that the bacteria gradually grew during 8–40 h, peaking at 32 h. After that, the protein content continued to accumulate in a fluctuating manner, reaching a maximum of approximately 1353.79 ± 24.63 μg/mL at 36 h. Consistent with the results from shake flask experiments, there was a lag in protein accumulation compared to biomass growth. The scale-up cultivation of NDS proved feasible and achieved higher biomass and protein yields compared to shake flask cultures. Based on the observed pattern of protein accumulation, the sampling time for subsequent optimization experiments was set between 32 and 40 h, a period during which the accumulation of the target product was higher.

#### 3.4.2. Effect of pH

Scale-up experiments in a 10 L bioreactor revealed critical pH-dependent growth characteristics of *R. marisflavi* NDS. A comparative analysis between pH-controlled (maintained at pH 7.3) and uncontrolled (allowed to fluctuate freely) fermentations demonstrated striking differences in culture performance (Figure 5A,B). Interestingly, the uncontrolled pH condition resulted in superior biomass accumulation and protein production compared to the tightly regulated system. These findings suggest that the dynamic pH environment better accommodates the changing metabolic requirements of *R. marisflavi* NDS throughout its growth cycle than static pH maintenance.

#### 3.4.3. Effect of Agitator Speed

As an aerobic fermentation process, protein production by *R. marisflavi* NDS is highly dependent on oxygen transfer efficiency, which is directly influenced by agitator speed. To optimize this parameter, we evaluated three agitation strategies: (i) constant 150 rpm, (ii) constant 180 rpm, and (iii) a two-phase variable speed protocol (150 rpm for 0–20 h, then 180 rpm). As shown in Figure 6, a constant agitation speed of 150 rpm supported optimal protein yield, whereas 180 rpm reduced protein production. Notably, the two-phase approach accelerated the attainment of peak biomass and protein levels while maintaining yields comparable to the 150 rpm condition. Furthermore, this strategy sustained higher protein accumulation than continuous 180 rpm agitation. Based on these results, we selected a two-phase protocol: an initial phase at 150 rpm for 20 h, followed by 180 rpm for 12 h, as the agitation strategy.

### 3.5. Analysis of Nutritional Composition of R. marisflavi NDS

*R. marisflavi* NDS was cultured under optimized fermentation conditions, and the biomass was harvested by centrifugation for nutritional composition analysis. As shown in Table 1, proximate composition analysis revealed a wet weight (WW) content of 79.1900 ± 0.0216% moisture, 15.6013 ± 0.0082% crude protein, 0.1023 ± 0.0026% crude lipid, 2.6997 ± 0.0021% carbohydrates, and 2.3767 ± 0.0205% ash. Meanwhile, dry weight (DW) equivalents of these components are provided for comparative analysis in Table 1.

Amino acid profiling indicated that the strain contained all essential and non-essential amino acids (Table 2). Among the essential amino acids, lysine was most abundant (1.3617 ± 0.0039%, WW), followed by leucine (0.8917 ± 0.0012%), valine (0.6713 ± 0.0009%), and isoleucine (0.6293 ± 0.0041%), while methionine showed the lowest concentration (0.2820 ± 0.0036%). In terms of non-essential amino acids, glutamate (2.5110 ± 0.0086%), alanine (1.1717 ± 0.0024%), aspartate (1.1390 ± 0.0008%), and proline (1.0910 ± 0.0119%) were present in relatively high concentrations, whereas cystine (0.0913 ± 0.0026%) was the least abundant. Meanwhile, DW equivalents of these components are provided for comparative analysis in Table 2. Notably, the total amino acid content (67.6238 ± 0.1050% DW) showed good correspondence with the crude protein content (75.0064 ± 0.0395% DW; Table 1), suggesting minimal protein degradation during processing. This alignment between amino acid quantification and traditional protein assays validates the reliability of our compositional analyses.

Fatty acid analysis identified nine distinct compounds, with monounsaturated fatty acids (MUFAs) constituting the predominant class (196.6575 ± 0.2690 mg/100 g WW; Table 3). The MUFA C14:1n-5 accounted for the majority of this fraction (141.9750 ± 0.2420 mg/100 g WW). Saturated fatty acids (SFAs) represented the second most prevalent class (21.9615 ± 0.0270 mg/100 g WW), primarily composed of C16:0 at 14.2405 ± 0.0255 mg/100 g WW. While polyunsaturated fatty acids (PUFAs) were the least abundant class (9.5550 ± 0.0110 mg/100 g WW), with C18:2n6c as the amin representative (8.5200 ± 0.0010 mg/100 g WW). Meanwhile, DW equivalents of these components are provided for comparative analysis in Table 3.

## 4. Discussion

At present, proteins represent one of the most costly and limiting components in animal feed formulations [1]. The development of alternative protein sources has become imperative to address the escalating demand accompanying the growth of worldwide livestock and aquaculture production. In recent decades, microbial protein, produced through the cultivation of bacteria, yeast, fungi, or microalgae, has emerged as a nutritionally viable and environmentally sustainable protein supplement for animal feed applications [1,28]. Microbial-derived proteins typically exhibit substantial protein content, constituting 30–70% of cellular dry mass across various microbial species [29]. Yeasts remain the most widely accepted and utilized source of single-cell protein in industrial applications [30]. Meanwhile, *Bacillus* species have gained considerable research interest for their unique capacity to valorize agricultural waste into high-quality protein feed through sustainable bioconversion processes [31,32]. Notably, protein content varies considerably among microbial species and strains, making comparative evaluation essential for identifying promising candidates. In this context, the newly isolated strain *R. marisflavi* NDS demonstrates considerable potential. As summarized in Table 4, its single-cell protein content (Table 1) is competitively high when compared with other well-established microbial protein producers—including fungi, bacteria, microalgae, and archaea—highlighting its suitability for further development in animal feed applications.

Optimal microbial proliferation and metabolic activity critically depend on carefully formulated growth medium and precisely controlled fermentation parameters, such as fermentation time, temperature, initial pH, and agitation rate [42]. In this study, various fermentation parameters and medium compositions were assessed for their impact on the growth and protein accumulation of the *R. marisflavi* NDS. The results indicated that the use of BPM supplemented with corn flour as a carbon source and KCl as an inorganic salt effectively enhanced protein content. This medium formulation is relatively simple, cost-effective, and easily scalable for industrial applications. The optimal fermentation conditions were determined to be as follows: a salinity of 20‰ NaCl, a temperature of 32 °C, an initial pH of 7.3, a fermentation duration of 22 h in flasks and 36 h in bioreactor, an inoculation volume of 1% (*v*/*v*), and a two-stage agitation strategy (150 rpm for the initial 20 h followed by 180 rpm for the remaining 12 h), with pH not controlled during the process (Table 5). This easily scalable and cost-effective process offers significant advantages for the large-scale application of this strain in feed protein supplementation.

Firstly, microbial growth and product synthesis are highly time-dependent during fermentation, with notable variations across different culture systems (e.g., shake flasks and bioreactors). Prolonged fermentation may lead to nutrient depletion, while the accumulation of byproducts could exert cytotoxic effects [43]. Therefore, this study systematically investigated the growth kinetics and protein accumulation patterns of the strain under both shake flask and bioreactor conditions. Results demonstrated that the NDS strain exhibits rapid growth (14 h under shake-flask and 32 h in the bioreactor), laying a foundation for its potential applications.

Secondly, *R. marisflavi* NDS was isolated from seawater and exhibits a high protein content, as indicated by preliminary study [24]. Salinity is a key environmental factor that affects microbial growth and metabolic activities. Elevated salinity can lead to protein denaturation, macromolecular damage, membrane integrity disruption, and impairment of biochemical pathways, ultimately suppressing microbial activity [44]. Therefore, this study investigated the optimal salinity using NaCl (the major salt component of seawater) for growth and protein accumulation in *R. marisflavi* NDS. The results demonstrated that the optimal NaCl concentration for its growth and protein synthesis was 20‰, with significant inhibition observed at higher salinity levels (Figure 2A), indicating that this bacterium is a moderate salt-tolerant strain. Additionally, temperature and initial pH are critical parameters, as deviations from their optimum ranges can adversely affect cell growth and biomass production. Substantial evidence has demonstrated that fermentation temperature plays a vital role in metabolite synthesis, enzymatic regulation, and target product biosynthesis during the mid-to-late fermentation phases [45]. *R. marisflavi* NDS exhibit optimal growth and protein accumulation at 32 °C.

Thirdly, a carbon source serves as the primary provider of energy and carbon skeleton for microbial growth and metabolism [46]. The preference for corn flour as the optimal carbon source has been documented in other Bacillus species. For example, *Bacillus subtilis* exhibited maximal neutral protease production when cultivated with corn flour as the primary carbon source [47]. Intriguingly, the growth of *R. marisflavi* NDS was significantly inhibited by common monosaccharides, disaccharides, organic acids, and alcohols, likely due to its inability to directly metabolize mono- or disaccharides. This suggests that its carbon metabolic network is primarily adapted for degrading complex polysaccharides (e.g., corn flour) rather than simple sugars. Similar findings have been reported in *Streptomyces* and *Caldicellulosiruptor bescii* [11,48]. These microorganisms typically possess specialized enzyme systems for complex polysaccharide degradation but lack direct metabolic pathways for simple sugars [49]. Nevertheless, we acknowledge that the tested concentrations (1% methanol/ethanol/organic acids) may have induced non-specific stress rather than reflecting a true lack of metabolic capacity. Therefore, future studies are required to define the optimal concentration for each carbon source.

Fourthly, in addition to carbon and nitrogen sources, inorganic ions play a crucial role in optimizing fermentation processes. For instance, in *Yarrowia lipolytica* fermentations for malonic acid production, optimal concentrations of Mn^2+^ and Zn^2+^ significantly increase product yield [50]. These essential metal ions function as cofactors for key enzymes while simultaneously maintaining cellular osmotic balance, thereby enhancing both fermentation efficiency and final product titer through multiple regulatory mechanisms [51]. However, in the present study, K^+^ was demonstrated to be the most critical ion for the growth of *R. marisflavi* NDS. This finding highlights the species-specific requirements for inorganic ions in bacterial fermentation.

Fifthly, agitation speed is a critical parameter that affects dissolved oxygen levels during fermentation [45,52]. Our findings revealed that higher agitation speeds negatively impacted protein accumulation. While a two-stage agitation control strategy did not increase the final protein yield, it did shorten the fermentation time required to reach peak protein levels. These results may be attributed to the relatively high aeration rate employed in this study, which ensured sufficient oxygen supply even at lower agitation speed. Furthermore, the observed enhancement in protein accumulation under unregulated pH conditions (Figure 5) implies that natural pH fluctuations may synchronize with the strain’s inherent metabolic cycles. This phenomenon could potentially reduce the energy demands associated with pH control in industrial-scale fermentation.

Admittedly, the single-factor approach employed here does not account for potential interactions among variables. However, it remains a widely used and pragmatically valid first step for initial screening and process baseline establishment, especially with novel or underexplored microbial strains [53]. This method allows for efficient identification of key influencing factors within a wide experimental space and has been effectively applied in numerous fermentation optimization studies [54,55]. The conditions derived from these experiments provided a robust starting point for scale-up, as evidenced by the successful translation to bioreactor-level protein production. Nevertheless, it is acknowledged that the single-factor design cannot discern interaction effects between parameters such as salinity, ion concentration, pH, and temperature. Therefore, while the high protein yield obtained in this study confirms the potential of *R. marisflavi* NDS as a promising single-cell protein producer, further optimization using statistical experimental designs, such as response surface methodology (RSM) or fractional factorial designs, will be necessary to elucidate interactions between key parameters and refine the cultivation process toward a global optimum [56,57].

Finally, under the optimized fermentation conditions, proximate analysis demonstrated a high protein content (75.0064 ± 0.0395% DW). This represents a substantial improvement over the previously reported value of 42.37% [24], reflecting the successful optimization of the fermentation process. Amino acid profiling demonstrated that this strain produces proteins rich in both essential and functional amino acids. Notably, it contains elevated levels of lysine (1.3617 ± 0.0039 WW)-a typically limiting amino acid in plant-based feeds-and glutamate (2.5110 ± 0.0086 WW), which enhances palatability. This complete amino acid profile, combined with the strain’s high protein yield, positions it as an ideal alternative protein source for animal feed formulations. Fatty acid composition revealed that *R. marisflavi* NDS biomass contains a lipid profile dominated by MUFAs (196.6575 ± 0.2690 mg/100 g), with the unusual C14:1n5 representing the predominant component (141.9750 ± 0.2420 mg/100 g). This distinctive lipid composition suggests enhanced feed energy density and digestibility. While these results provide foundational nutritional characterization, future studies should address nucleic acid content, mineral speciation, and contaminant screening to fully validate its dietary applications.

## 5. Conclusions

In this study, the systematic optimization of *R. marisflavi* NDS fermentation demonstrates its high potential as a sustainable microbial protein source for industrial applications. The established optimal fermentation conditions were: salinity of 20‰ NaCl, temperature at 32 °C, an initial pH of 7.3 (allowing for natural pH drift during cultivation), an agitation rate at 150 rpm for the initial 20 h followed by 180 rpm for the remaining 12 h, and a medium comprising 1% corn flour, 1% peptone, 0.3% beef extract, and 0.2% KCl. Under these optimized conditions in a 10 L bioreactor, the biomass composition was determined to be 2.3767 ± 0.0205% crude ash, 15.6013 ± 0.0082% protein, 0.1023 ± 0.0026% lipid, and 2.6997 ± 0.0021% carbohydrates (WW). Amino acid profiling revealed that this strain produces diverse proteins rich in essential amino acids, with a notably high lysine content (1.3617 ± 0.0039%, WW), underscoring its excellent nutritional value. Collectively, *R. marisflavi* NDS exhibits three key advantages: (1) rapid growth with tolerance to 20‰ salinity of NaCl; (2) high-efficiency production on low-cost substrates like corn flour; and (3) an energy-saving fermentation process that does not require precise pH control. These superior traits position it as an ideal microbial protein candidate for animal feed. Future research will focus on multifactorial optimization of fermentation conditions, pilot-scale validation, agricultural waste valorization, and a comprehensive life cycle assessment to facilitate industrial-scale application.

## Figures and Tables

**Figure 1 foods-14-03066-f001:**
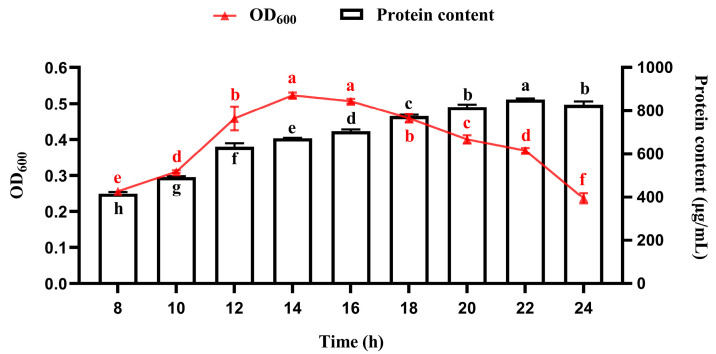
Growth and protein accumulation profile of *R. marisflavi* NDS during fermentation. Growth was monitored by measuring optical density at OD_600_, and protein concentration was determined by the BCA assay. Data points represent the mean ± SD of three independent replicates (*n* = 3). Mean values labeled with different letter of the same color are significantly different (*p* < 0.05).

**Figure 2 foods-14-03066-f002:**
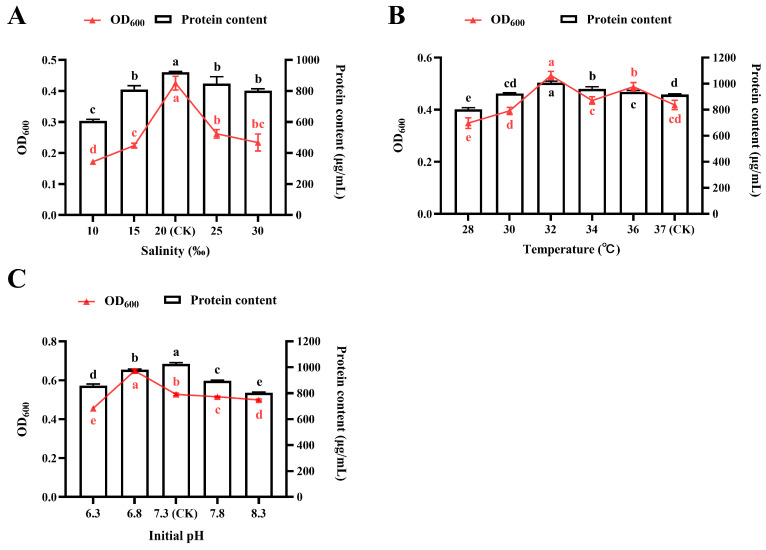
Effects of cultivation conditions on the growth and protein accumulation of *R. marisflavi* NDS. (**A**), NaCl salinity; (**B**), temperature; (**C**), initial pH. Growth was measured by optical density at OD_600_, and protein concentration was quantified by the BCA assay. Data points represent the mean SD of three independent biological replicates (*n* = 3). For each panel, mean values labeled with different letter of the same color are significantly different (*p* < 0.05). CK, control check.

**Figure 3 foods-14-03066-f003:**
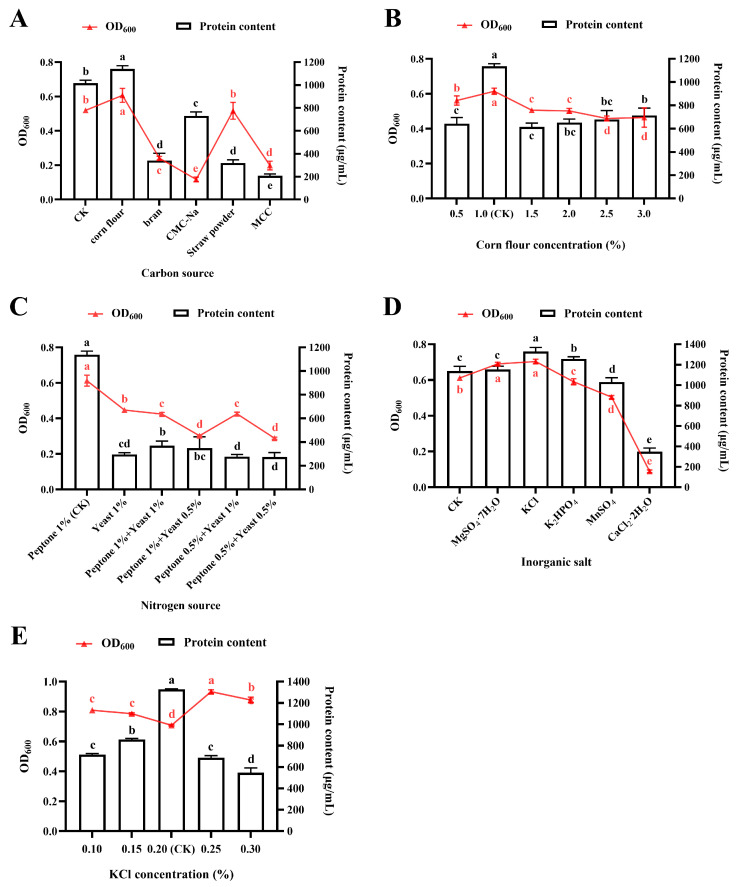
Effect of medium composition on the growth and protein production of *R. marisflavi* NDS. (**A**) Carbon source (1.0% *w*/*v* or *v*/*v*), (**B**) corn flour concentration (% *w*/*v*), (**C**) nitrogen source, (**D**) inorganic salt (0.2% *w*/*v*), and (**E**) KCl concentration (% *w*/*v*). Growth was monitored by measuring optical density at OD_600_, and protein concentration was determined by the BCA assay. Data represent the mean SD of three independent biological replicates (*n* = 3). For each panel, mean values labeled with different letter of the same color are significantly different (*p* < 0.05). CK, control check; CMC-Na, carboxymethylcellulose sodium; MCC, microcrystalline cellulose.

**Figure 4 foods-14-03066-f004:**
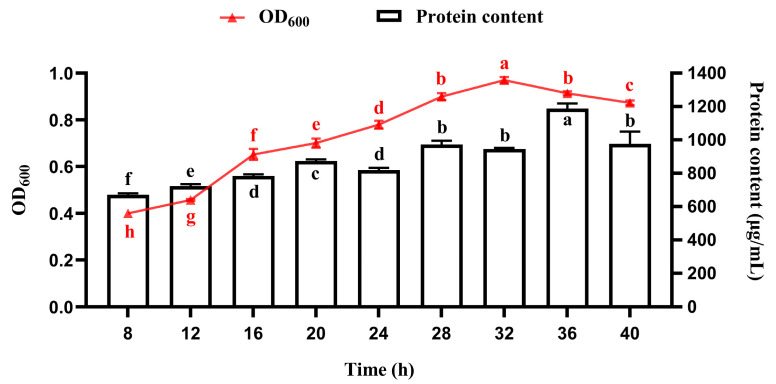
Growth and protein production profile of *R. marisflavi* NDS in a 10 L bioreactor. Growth was monitored by measuring optical density at OD_600_, and protein concentration was determined by the BCA assay. Data points represent the mean ± SD of three independent replicates (*n* = 3). Mean values labeled with different letter of the same color are significantly different (*p* < 0.05).

**Figure 5 foods-14-03066-f005:**
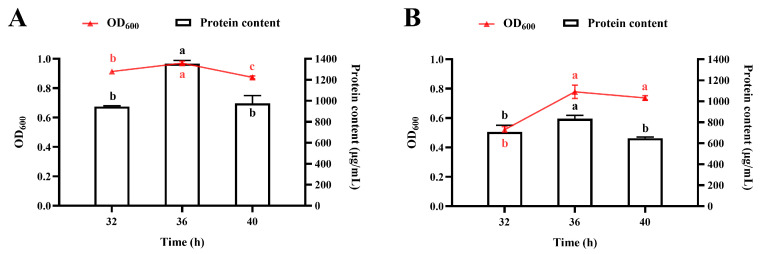
Effect of pH on growth and protein accumulation in *R. marisflavi* NDS cultured in a 10 L bioreactor. (**A**), uncontrolled pH; (**B**), pH-controlled (maintained at 7.30). Growth was monitored by measuring optical density at OD_600_, and protein concentration was determined by the BCA assay. Data points represent the mean ± SD of three independent replicates (*n* = 3). For each panel, mean values labeled with different letter of the same color are significantly different (*p* < 0.05).

**Figure 6 foods-14-03066-f006:**
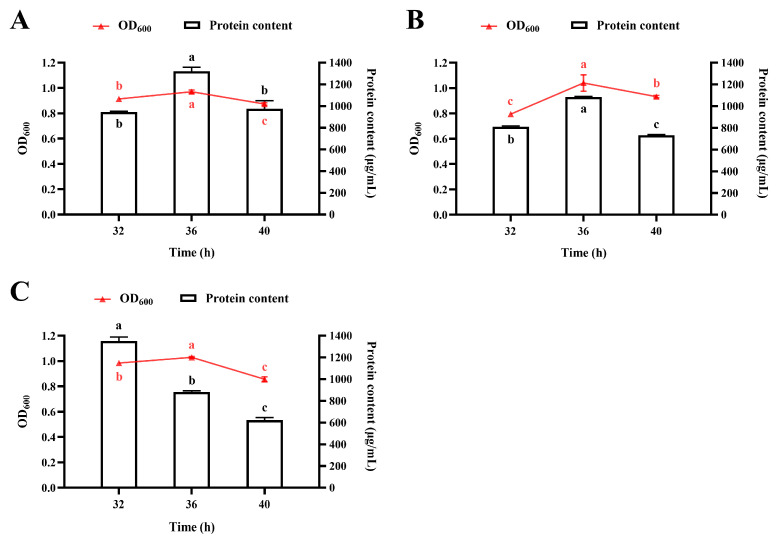
Effect of agitator speed on growth and protein accumulation in *R. marisflavi* NDS cultured in a 10 L bioreactor. (**A**), constant 150 rpm; (**B**), constant 180 rpm; (**C**), two-phase agitation (150 rpm for 0–20 h, then 180 rpm). Growth was monitored by measuring optical density at OD_600_, and protein concentration was determined by the BCA assay. Data points represent the mean ± SD of three independent replicates (*n* = 3). For each panel, mean values labeled with a different letter of the same color are significantly different (*p* < 0.05).

**Table 1 foods-14-03066-t001:** Proximate composition of *R. marisflavi* NDS biomass cultivated under optimized fermentation conditions.

Proximate Composition	Content (Wet Weight, WW)	Content (Dry Weight, DW)	Unit
Moisture	79.1900 ± 0.0216	0	%
Crude ash	2.3767 ± 0.0205	11.4263 ± 0.0988	%
Crude protein	15.6013 ± 0.0082	75.0064 ± 0.0395	%
Crude lipid	0.1023 ± 0.0026	0.4920 ± 0.0126	%
Carbohydrates	2.6997 ± 0.0021	12.9792 ± 0.0099	%

Data represent mean ± SD of three biological replicates (*n* = 3).

**Table 2 foods-14-03066-t002:** Amino acid composition of *R. marisflavi* NDS biomass cultivated under optimized fermentation conditions.

Amino Acid Types	Content (Wet Weight, WW)	Content (Dry Weight, DW)	Unit
Lysine	1.3617 ± 0.0039	6.5472 ± 0.0193	%
Leucine	0.8917 ± 0.0012	4.2869 ± 0.0066	%
Valine	0.6713 ± 0.0009	3.2283 ± 0.0051	%
Isoleucine	0.6293 ± 0.0041	3.0259 ± 0.0208	%
Threonine	0.5530 ± 0.0022	2.6582 ± 0.0102	%
Phenylalanine	0.5123 ± 0.0040	2.4639 ± 0.0189	%
Tryptophan	0.5127 ± 0.0084	2.4656 ± 0.0396	%
Histidine	0.4813 ± 0.0082	2.3110 ± 0.0413	%
Methionine	0.2820 ± 0.0036	1.3556 ± 0.0183	%
Glutamic acid	2.5110 ± 0.0086	12.0731 ± 0.0419	%
Alanine	1.1717 ± 0.0024	5.6330 ± 0.0110	%
Aspartic acid	1.1390 ± 0.0008	5.4755 ± 0.0038	%
Proline	1.0910 ± 0.0119	5.2446 ± 0.0560	%
Glycine	0.6677 ± 0.0026	3.2086 ± 0.0134	%
Arginine	0.5813 ± 0.0012	2.7943 ± 0.0060	%
Tyrosine	0.4680 ± 0.0051	2.2506 ± 0.0248	%
Serine	0.4497 ± 0.0021	2.1625 ± 0.0094	%
Cystine	0.0913 ± 0.0026	0.4390 ± 0.0119	%
Total amino acids	14.0660 ± 0.0218	67.6238 ± 0.1050	%

Data represent mean ± SD of three biological replicates (*n* = 3).

**Table 3 foods-14-03066-t003:** Fatty acid composition of *R. marisflavi* NDS biomass cultivated under optimized fermentation conditions.

Fatty Acid Types	Content (Wet Weight, WW)	Content (Dry Weight, DW)	Unit
C14:0	3.7375 ± 0.0065	17.9247 ± 0.0261	mg/100 g
C14:1n5	141.9750 ± 0.2420	682.6122 ± 0.9517	mg/100 g
C15:0	2.3615 ± 0.0055	11.3510 ± 0.0219	mg/100 g
C15:1n5	49.0535 ± 0.1175	235.9087 ± 0.4731	mg/100 g
C16:0	14.2405 ± 0.0255	68.3990 ± 0.1358	mg/100 g
C18:0	1.6220 ± 0.0030	7.7949 ± 0.0126	mg/100 g
C18:1n9c	5.6290 ± 0.0020	27.0641 ± 0.0785	mg/100 g
C18:2n6c	8.5200 ± 0.0010	40.9615 ± 0.0039	mg/100 g
C18:3n3	1.0350 ± 0.0110	4.9679 ± 0.0446	mg/100 g
SFAs	21.9615 ± 0.0270	105.4696 ± 0.1406	mg/100 g
MUFAs	196.6575 ± 0.2690	945.5850 ± 1.0657	mg/100 g
PUFAs	9.5550 ± 0.0110	45.9294 ± 0.0448	mg/100 g
TFAs	228.1740 ± 0.2706	1096.9840 ± 1.0758	mg/100 g

Data represent mean ± SD of three biological replicates (*n* = 3). SFAs, saturated fatty acids; MUFAs, monounsaturated fatty acids; PUFAs, polyunsaturated fatty acids. TFAs, total fatty acids.

**Table 4 foods-14-03066-t004:** Comparison of the single cell protein content of *R. marisflavi* NDS with other strains and species used as single cell proteins.

Microorganism	Content (DW)	Unit	References
Fungi			
*Pleurotus florida*	63	%	[33]
*Aspergillus niger*	17–50	%	[33]
*Saccharomyces cerevisiae*	24–50	%	[34]
*Yarrowia lipolytica*	48–54	%	[34]
*Kluyveromyces marxianus*	59	%	[35]
*Geotrichum candidum*	40	%	[36]
Bacteria			
*Rhodopseudomonas palustris*	65	%	[37]
*Methylococcus capsulatus*	70	%	[38]
*Methylomonas* sp. DM580	76	%	[39]
*Rhodopseudomonas faecalis* PA2	62.7	%	[40]
*Aifella marina* STW181	46.4	%	[40]
Microalgae			
*Galdieria sulphuraria*	44	%	[36]
*Nannochloris* sp.	31–68	%	[41]
*Phaeodactylum tricornutum*	18–57	%	[41]
Archaea			
*Haloarcula* sp. IRU1	76	%	[35]

**Table 5 foods-14-03066-t005:** Summary of optimized fermentation conditions for enhanced sing-cell protein production by *R. marisflavi* NDS.

Parameter	Optimized Condition
Salinity	20‰ NaCl
Temperature	32 °C
Initial pH	7.3
Medium composition	1% (*w*/*v*) corn flour, 1% peptone 0.3% beef extract, 0.2% KCl
Fermentation duration	22 h (Flask)/32 h (10 L bioreactor, total time)
pH control	Uncontrolled (allowed to fluctuate freely)
Agitator speed	150 rpm for initial 20 h, then 180 rpm for the remaining 12 h (10 L bioreactor)

## Data Availability

The original contributions presented in the study are included in the article/Appendix A, further inquiries can be directed to the corresponding authors.

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
