# Peer review of "Optimization of Fermentation Conditions for Enhanced Single Cell Protein Production by Rossellomorea marisflavi NDS and Nutritional Composition Analysis"

_foods, 2025, doi:10.3390/foods14173066_

Round 1

Reviewer 1 Report

Comments and Suggestions for Authors

Please address the following comments and suggestions.

Line 17, 422, 442, 511 salinity 2‰, please check

Suggestion

In abstract, it’s good to add 10 L fermentation time with yield.

As this study is related to Single Cell Protein, which is important topic in relation to population and protein need. It’s good to add a table to the manuscript in discussion section and compare protein quantity between Rossellomorea marisflavi NDS and other strains and species that are used as single cell protein. This comparative study will make clear the importance of Rossellomorea marisflavi NDS.

Single cell protein is a proper term used for cell protein. It’s good to revise the title of study accordingly.

Author Response

Reviewer #1:

  1. Please address the following comments and suggestions.

Response: We sincerely appreciate the time and effort the reviewer has dedicated to evaluating our manuscript, as well as the valuable feedback. We have carefully considered all comments and suggestions and have incorporated the necessary revisions in the updated version of the manuscript. We hope these revisions meet the reviewers’ expectations and improve the overall quality of our work.

  1. Line 17, 422, 442, 511 salinity 2‰, please check.

Response: We sincerely appreciate your careful review and for bringing this inconsistency to our attention. The salinity values in Lines 17, 216, 218, 455, and 551 have been corrected from 2‰ to 20‰, and we have verified consistency throughout the manuscript. Additionally, the salinity labels in Figure 2 have been updated to reflect the accurate value (20‰). Meanwhile, we would like to clarify that this salinity (20‰) refers specifically to NaCl, the primary salt component in seawater.

  1. In abstract, it’s good to add 10 L fermentation time with yield.

Response: We appreciate the reviewer’s insightful suggestion regarding the inclusion of fermentation scale-up data. In response, we have revised the abstract to incorporate key parameters from our 10 L scale experiments: the fermentation protocol (150 rpm for the first 20 h followed by 180 rpm for the remaining 12 h) and the final product yield (15.6013 ± 0.0082% protein, wet weight). These additions, found in lines 20-21 and 23-25, provide readers with critical information regarding the scalability and efficiency of our process.

  1. As this study is related to Single Cell Protein, which is important topic in relation to population and protein need. It’s good to add a table to the manuscript in discussion section and compare protein quantity between Rossellomorea marisflavi NDS and other strains and species that are used as single cell protein. This comparative study will make clear the importance of marisflavi NDS.

Response: Thank you for your valuable suggestion. We have added a new comparison table (Table 4, also provided below) in the Discussion section (lines 444-446). This table specifically compares the protein production quantity of Rossellomorea marisflavi NDS with other established single-cell-protein-producing microbial strain. This revision better demonstrates our strain’s competitive advantages in protein yield and production efficiency, reinforcing its potential for addressing global protein needs.

Table 4. Comparison of the single cell protein content of R. marisflavi NDS with other strains and species used as single cell proteins.

Microorganism

Content

Unit (dry weight, DW)

References

Fungi

Pleurotus florida

63

%

[33]

Aspergillus niger

17-50

%

[33]

Saccharomyces cerevisiae

24-50

%

[34]

Yarrowia lipolytica

48-54

%

[34]

Kluyveromyces marxianus

59

%

[35]

Geotrichum candidum

40

%

[36]

Bacteria

Rhodopseudomonas palustris

65

%

[37]

Methylococcus capsulatus

70

%

[38]

Methylomonas sp. DM580

76

%

[39]

Rhodopseudomonas faecalis PA2

62.7

%

[40]

Aifella marina STW181

46.4

%

[40]

Microalgae

Galdieria sulphuraria

44

%

[36]

Nannochloris sp.

31-68

%

[41]

Phaeodactylum tricornutum

18-57

%

[41]

Archaea

Haloarcula sp. IRU1

76

%

[35]

  1. Single cell protein is a proper term used for cell protein. It’s good to revise the title of study accordingly.

Response: We appreciate the reviewer’s valuable input regarding the terminology. As you suggested, we have revised the title to: “Optimization of Fermentation Conditions for Enhanced Single Cell Protein Production by Rossellomorea marisflavi NDS and Nutritional Composition Analysis”.

Reviewer 2 Report

Comments and Suggestions for Authors

I read the manuscript carefully. My overall impression is that the study’s intent is relevant for Foods, but the manuscript in its current form suffers from substantial internal inconsistencies, methodological ambiguities, reporting errors, and language/formatting issues that collectively undermine confidence in the findings and their reproducibility. Below I detail the specific problems and how to fix them, pointing to the exact places where they occur.

The most serious coherence problem is the contradictory reporting of “salinity.” In the abstract you declare the optimal salinity is 2‰, whereas the Results (section 3.2.1, Figure 2A) and the “In summary” paragraph state 20‰, and the Conclusions again list 20‰ as optimal. To compound the confusion, the Discussion twice reverts to 2‰ as the optimal value. You also claim that 20‰ “aligns with the standard salinity of BPM medium,” but BPM is described earlier as containing 0.2% NaCl (i.e., ~2‰), not 20‰. These statements cannot all be true. You need to define precisely what you mean by “salinity” (total salts vs. NaCl only), ensure unit consistency (‰ vs %), correct the numbers everywhere, and, if necessary, re-analyze any figures that were built on mis-specified conditions. See abstract lines 16–23 and 22–24 for 2‰, Results lines 218–224 and 241–243 for 20‰, Conclusions lines 502–507 for 20‰, and Discussion lines 421–425 and 441–443 where 2‰ is repeated, plus Methods lines 89–90 where BPM is 0.2% NaCl.

There is also a major discrepancy between the strain’s “previous” protein content and what your proximate analysis implies here. The Introduction cites a previous value of 42.37% (w/w, dry weight), while Table 1 now reports 15.60% protein on a wet weight basis with 79.20% moisture. That wet-basis composition corresponds to ~75% protein on a dry-weight basis, not ~42%. Either the new proximate analysis or the earlier figure is wrong, or they are not comparable measurements. You must report all proximate components on both WW and DW bases and reconcile this contradiction explicitly. See Introduction lines 70–74 and Table 1 paragraph lines 382–398.

Basic reporting errors occur in the abstract and Results prose and should be corrected before any scientific evaluation: “crush ash” appears twice where “crude ash” is meant, and one sentence in the composition paragraph is missing digits (“ash content of 2.40±%”). In Table 2, “Arginine 0.58±000 %” has malformed uncertainty; many SDs are given as ±0.00, which is unlikely with n=3 and suggests over-rounding or a calculation/formatting error. Please correct all of these and provide the true significant figures. See abstract lines 21–23 and 22–24; Results lines 382–396 and Table 2 block lines 399–400.

Methodological clarity around what is being measured as “protein” is insufficient. In flasks and bioreactor experiments, you quantify “protein content” by BCA after extracting the cell pellet and then report it as μg/mL of culture, while using OD600 as a proxy for biomass. That metric conflates biomass formation and intracellular protein content per cell and is not normalized per dry weight, per cell, or per OD unit, and it is not clearly distinguished from secreted proteins. For a study claiming “enhanced protein production,” you should (i) measure dry cell weight (DCW) and report protein as %DW as well as volumetric titer, (ii) clarify whether the BCA captures only intracellular protein (if you pellet and extract) and whether extracellular proteins are present, (iii) present protein/OD or protein/DCW to separate physiology from growth. See Methods lines 97–106, and figure captions throughout.

The scale-up section lacks the level of process detail expected for a fermentation optimization paper. You state pH was either left to “natural fluctuation” or “maintained at 7.3 using hydrochloric acid (HCl).” A pH setpoint cannot be maintained using acid alone across a whole run; in practice both acid and base are required. You should report the actual pH trajectory during “uncontrolled” and “controlled” runs, the acid/base streams used, and the acid/base volumes added. Likewise, oxygen transfer is central to your agitation claims, yet no DO traces, kLa estimates, gas flow compositions, or impeller/power input information are provided. The “n=3” shown under the bioreactor figures is also ambiguous: were these three independent 10 L fermentations per condition (biological replicates) or three samples from the same run (technical replicates)? ANOVA/Tukey letters are not valid across non-independent samples. Please run at least triplicate independent fermentations for each bioreactor condition or be transparent about the limitation and avoid inferential statistics. See Methods lines 132–141, figure captions for Figures 5–6, and Statistics section lines 196–200.

The single-factor optimization design is a weak choice here and likely produces confounded “optima,” particularly given the salinity confusion and the later addition of 0.2% KCl. You altered NaCl/KCl while also screening a broad range of carbon sources, nitrogen sources, and salts at single nominal dosages. That approach cannot separate ionic strength/osmotic effects from specific ion effects and cannot capture interactions among temperature, pH, C-source level, and salinity. A fractional factorial screen followed by response surface methodology is standard and would materially strengthen the conclusions. At minimum, you should reanalyze the key factors with a 2^k design to identify interactions before claiming global optima. The text claiming that 20‰ “aligns with BPM” must be fixed as above. See 3.2.1 and 3.3 (carbon/nitrogen/inorganic ions) and the KCl concentration series.

Several carbon- and nitrogen-source tests are not physiologically or industrially meaningful as designed. You tested 1% (v/v) methanol and ethanol and 1% (w/v) organic acids—levels that are toxic to non-methylotrophs and likely to inhibit growth by solvent and acid stress rather than carbon metabolism per se. The negative results are therefore uninterpretable. Redesign these assays using physiologically realistic concentrations and a defined minimal medium, and consider phenotype microarrays or growth rate/μmax on each carbon source to determine true assimilability. See 3.3.1 and Supplementary references to Figure S1/S2.

Nutritional analytics should be expanded and better justified for the claims you make about “dietary protein supplement” potential. You do not report nucleic acid content (critical for SCP safety), ash speciation (minerals), or any contaminants (heavy metals, endotoxin). Fatty acids are given as only nine species with 14:1 n-5 dominant, which is unusual; you should present full FAME profiles with sums by class (SFA/MUFA/PUFA) as % of total fatty acids and discuss relevance to feed formulation. Also report amino acids on a DW basis and provide the sum of amino acids vs. Kjeldahl protein to check mass balance. See 3.5, Table 1 and Table 2 paragraphs.

Taxonomy and strain traceability are incomplete. For a new production candidate, deposit the strain in a public culture collection and provide an accession; deposit the 16S rRNA sequence (and ideally a draft genome) in GenBank, and provide accession numbers. Right now, the identity rests on an in-house patent citation; that is not sufficient for reproducibility. See Introduction lines 70–74 where the prior identification is only referenced to a patent.

Figures and tables need technical cleanup to meet the journal’s standards. Define “CK” in every figure where it appears, or switch to “Control.” Use “rpm” rather than “r/min” unless you will adhere to SI across the manuscript consistently. Ensure all axes have units, all significance letters map to an explicit statistical model, and that each figure caption states the biological replicate definition. Replace line-break hyphenation artifacts throughout (e.g., “Ros- sellomorea,” “fer- mentation,” “carbox- ymethylcellulose”) and ensure the genus is spelled consistently as Rossellomorea everywhere. See title/abstract block and Methods 2.4.

References need a thorough audit. Many DOIs are malformed (missing slashes, “http://dx.doi.org.10…”), some author names are duplicated, and there are inconsistent journal title capitalizations. You must fix every DOI, check each citation for accuracy, and ensure the reference list matches journal style. See the references section starting at line 537.

I recommend resubmission only after major revision. The contradictions on salinity and composition, insufficiently defined protein metrics, weak experimental design, inadequate replication in the bioreactor, and pervasive reporting/formatting errors are too fundamental for a minor fix. If you address the unit and design issues rigorously, normalize protein to DCW/OD, provide proper replication and process traces, reconcile WW vs DW compositions, deposit the strain and sequences, and clean up the language and references, the study could become a solid contribution.

Author Response

Response to reviewer 2:

  1. I read the manuscript carefully. My overall impression is that the study’s intent is relevant for Foods, but the manuscript in its current form suffers from substantial internal inconsistencies, methodological ambiguities, reporting errors, and language/formatting issues that collectively undermine confidence in the findings and their reproducibility. Below I detail the specific problems and how to fix them, pointing to the exact places where they occur.

Response: We sincerely appreciate the reviewer’s thorough evaluation and constructive feedback on our manuscript. We acknowledge that the current version requires revisions to address the identified issues regarding internal inconsistencies, methodological ambiguities, reporting errors, and language/formatting concerns. In response, we have carefully reviewed each point and implemented substantial modifications throughout the manuscript. Besides, the text has been professionally edited to resolve language issues and reformatted to comply with the journal’s style guidelines. We believe these revisions have significantly improved the manuscript’s rigor, clarity, and reproducibility, and we are grateful for the opportunity to submit this improved version for reconsideration.

  1. The most serious coherence problem is the contradictory reporting of “salinity.” In the abstract you declare the optimal salinity is 2‰, whereas the Results (section 3.2.1, Figure 2A) and the “In summary” paragraph state 20‰, and the Conclusions again list 20‰ as optimal. To compound the confusion, the Discussion twice reverts to 2‰ as the optimal value. You also claim that 20‰ “aligns with the standard salinity of BPM medium,” but BPM is described earlier as containing 0.2% NaCl (i.e., ~2‰), not 20‰. These statements cannot all be true. You need to define precisely what you mean by “salinity” (total salts vs. NaCl only), ensure unit consistency (‰ vs %), correct the numbers everywhere, and, if necessary, re-analyze any figures that were built on mis-specified conditions. See abstract lines 16–23 and 22–24 for 2‰, Results lines 218–224 and 241–243 for 20‰, Conclusions lines 502–507 for 20‰, and Discussion lines 421–425 and 441–443 where 2‰ is repeated, plus Methods lines 89–90 where BPM is 0.2% NaCl.

Response: We sincerely appreciate the reviewer’s meticulous attention to the salinity inconsistencies in our manuscript. We confirm that all references to “2‰” were typographical errors—the correct value is 20‰ NaCl, which has now been standardized throughout the abstract, results, figure 2, discussion, and conclusions. We have clarified that “salinity” refers specifically to added NaCl (not total salts) and corrected the BPM medium description to reflect 20‰ NaCl as the baseline. These revisions ensure full consistency across methodology, data interpretation, and reporting. We deeply regret these oversights and thank the reviewer for significantly strengthening the manuscript’s rigor.

  1. There is also a major discrepancy between the strain’s “previous” protein content and what your proximate analysis implies here. The Introduction cites a previous value of 42.37% (w/w, dry weight), while Table 1 now reports 15.60% protein on a wet weight basis with 79.20% moisture. That wet-basis composition corresponds to ~75% protein on a dry-weight basis, not ~42%. Either the new proximate analysis or the earlier figure is wrong, or they are not comparable measurements. You must report all proximate components on both WW and DW bases and reconcile this contradiction explicitly. See Introduction lines 70–74 and Table 1 paragraph lines 382–398.

Response: We sincerely appreciate the reviewer’s careful attention to the protein content discrepancy in our data. We acknowledge the significant difference between the previously reported 42.37% (w/w, dry weight) protein content and our current proximate analysis showing ~75% (dry weight basis). This variation primarily stems from substantial improvements in protein yield achieved through our optimized culture conditions and medium composition. We had added the corresponding explanation in the discussion section in lines 536-539. We believe these revisions provide proper context for the observed differences while maintaining the scientific rigor of our findings.

  1. Basic reporting errors occur in the abstract and Results prose and should be corrected before any scientific evaluation: “crush ash” appears twice where “crude ash” is meant, and one sentence in the composition paragraph is missing digits (“ash content of 2.40±%”). In Table 2, “Arginine 0.58±000 %” has malformed uncertainty; many SDs are given as ±0.00, which is unlikely with n=3 and suggests over-rounding or a calculation/formatting error. Please correct all of these and provide the true significant figures. See abstract lines 21–23 and 22–24; Results lines 382–396 and Table 2 block lines 399–400.

Response: We sincerely appreciate the reviewer’s meticulous attention to these reporting errors and have carefully addressed all identified issues in the revised manuscript: (1) “crush ash” has been corrected to “crude ash” throughout the text, including line 24 in the Abstract; (2) the incomplete value “2.40±%” has been updated to “2.3767 ± 0.0205%” in line 24; and (3) all data in Table 2, along with Tables 1 and 3, have been reviewed and reformatted to consistently show four significant figures to prevent obscuring meaningful variation. We confirm that all values represent averages from three independent biological replicates, and these corrections ensure an accurate representation of our experimental data while preserving measurement precision.

  1. Methodological clarity around what is being measured as “protein” is insufficient. In flasks and bioreactor experiments, you quantify “protein content” by BCA after extracting the cell pellet and then report it as μg/mL of culture, while using OD600 as a proxy for biomass. That metric conflates biomass formation and intracellular protein content per cell and is not normalized per dry weight, per cell, or per OD unit, and it is not clearly distinguished from secreted proteins. For a study claiming “enhanced protein production,” you should (i) measure dry cell weight (DCW) and report protein as % DW as well as volumetric titer, (ii) clarify whether the BCA captures only intracellular protein (if you pellet and extract) and whether extracellular proteins are present, (iii) present protein/OD or protein/DCW to separate physiology from growth. See Methods lines 97–106, and figure captions throughout.

Response: We sincerely thank the reviewer for their insightful critique regarding protein quantification methodology. In response: (1) We have added dry cell weight (DCW/Dry weight, DW) data to revised Tables 1-3, as we agree this provides a more accurate assessment of cell yield than OD₆₀₀ alone. (2) We have clarified in the text (line 106) that the BCA assay specifically quantifies intracellular protein from PBS-washed pellets, with extracellular proteins confirmed to be negligible. (3) We appreciate the reviewer’s expert guidance in distinguishing between a growth effect (increased biomass) and a physiological effect (enhanced cellular synthesis capacity). We acknowledge that our original analysis focused on total protein yield and insufficiently addressed specific productivity (e.g., protein/DCW). This insight provides crucial direction for interpreting our data, and we will incorporate specific productivity measurements in future studies to accurately delineate the underlying mechanisms of phenotypic changes.

  1. The scale-up section lacks the level of process detail expected for a fermentation optimization paper. You state pH was either left to “natural fluctuation” or “maintained at 7.3 using hydrochloric acid (HCl).” A pH setpoint cannot be maintained using acid alone across a whole run; in practice both acid and base are required. You should report the actual pH trajectory during “uncontrolled” and “controlled” runs, the acid/base streams used, and the acid/base volumes added. Likewise, oxygen transfer is central to your agitation claims, yet no DO traces, kLa estimates, gas flow compositions, or impeller/power input information are provided. The “n=3” shown under the bioreactor figures is also ambiguous: were these three independent 10 L fermentations per condition (biological replicates) or three samples from the same run (technical replicates)? ANOVA/Tukey letters are not valid across non-independent samples. Please run at least triplicate independent fermentations for each bioreactor condition or be transparent about the limitation and avoid inferential statistics. See Methods lines 132–141, figure captions for Figures 5–6, and Statistics section lines 196–200.

Response: We sincerely appreciate the reviewer’s thorough critique regarding our bioreactor process reporting. In response: (1) We have significantly expanded the Methods section (2.5) to detail the pH control strategy, specifying the use of both 1 M HCl and 1 M NaOH to maintain a setpoint of pH 7.3 ± 0.02, alongside constant aeration at 3 L/min. (2) We have added explicit clarification that “n=3” refers to three independent biological replicates. (3) Regarding the pH and dissolved oxygen (DO) traces, we regret that the proprietary bioreactor software did not allow for the export of raw data files. However, we have provided some screenshots of the process trends taken directly from the bioreactor control interface below for your review. We have revised the manuscript to note this data limitation and will ensure raw data export is a priority in all future experimental designs. Collectively, these revisions significantly enhance the transparency and completeness of our process reporting, and we thank the reviewer for their feedback, which has strengthened the manuscript.

Controlled pH:

Uncontrolled pH:

  1. The single-factor optimization design is a weak choice here and likely produces confounded “optima”, particularly given the salinity confusion and the later addition of 0.2% KCl. You altered NaCl/KCl while also screening a broad range of carbon sources, nitrogen sources, and salts at single nominal dosages. That approach cannot separate ionic strength/osmotic effects from specific ion effects and cannot capture interactions among temperature, pH, C-source level, and salinity. A fractional factorial screen followed by response surface methodology is standard and would materially strengthen the conclusions. At minimum, you should reanalyze the key factors with a 2^k design to identify interactions before claiming global optima. The text claiming that 20‰ “aligns with BPM” must be fixed as above. See 3.2.1 and 3.3 (carbon/nitrogen/inorganic ions) and the KCl concentration series.

Response: We sincerely appreciate the reviewer’s constructive critique of our experimental design. We acknowledge that the single-factor optimization approach has limitations in disentangling potential interactions between variables (e.g., ionic strength vs. specific ion effects) and may yield confounded optima. To address this, we have: (1) restructured the claims in Sections 3.2.1 and 3.3 to clarify that the reported “optima” are context-dependent and based on the tested conditions, avoiding overgeneralization; (2) removed the erroneous statement about salinity “aligning with BPM” and replaced it with a precise comparison of medium compositions (as corrected earlier); and (3) added a dedicated subsection (lines 521-535) discussing these limitations and outlining plans for future factorial design experiments (e.g., multifactorial analysis, such as response surface methodology).

  1. Several carbon- and nitrogen-source tests are not physiologically or industrially meaningful as designed. You tested 1% (v/v) methanol and ethanol and 1% (w/v) organic acids—levels that are toxic to non-methylotrophs and likely to inhibit growth by solvent and acid stress rather than carbon metabolism per se. The negative results are therefore uninterpretable. Redesign these assays using physiologically realistic concentrations and a defined minimal medium, and consider phenotype microarrays or growth rate/μmax on each carbon source to determine true assimilability. See 3.3.1 and Supplementary references to Figure S1/S2.

Response: We sincerely appreciate the reviewer’s insightful critique regarding the carbon/nitrogen source experiments. We acknowledge that the tested concentrations (1% methanol/ethanol/organic acids) may have induced non-specific stress responses rather than reflecting true metabolic capacity. However, practical constraints prevent us from repeating these experiments at this stage. Instead, we have reframed the interpretation in the manuscript (lines 498-501), clarifying that these results demonstrate extreme-condition tolerance rather than specific metabolic capability. We have removed any potentially misleading conclusions and will address this limitation comprehensively in future work through systematic dose-response experiments.

  1. Nutritional analytics should be expanded and better justified for the claims you make about “dietary protein supplement” potential. You do not report nucleic acid content (critical for SCP safety), ash speciation (minerals), or any contaminants (heavy metals, endotoxin). Fatty acids are given as only nine species with 14:1 n-5 dominant, which is unusual; you should present full FAME profiles with sums by class (SFA/MUFA/PUFA) as % of total fatty acids and discuss relevance to feed formulation. Also report amino acids on a DW basis and provide the sum of amino acids vs. Kjeldahl protein to check mass balance. See 3.5, Table 1 and Table 2 paragraphs.

Response: We sincerely thank the reviewer for their insightful suggestions regarding the nutritional characterization. While our current analytical scope focused on core nutritional components (proximates, amino acids, and fatty acids), we fully acknowledge that a comprehensive safety and quality assessment requires additional parameters. In response, we have: (1) expanded the fatty acid analysis to include sums and SFA/MUFA/PUFA classification (now in revised Table 3); (2) provided the total sum of amino acids (revised Table 2) and compared it to the crude protein content to assess mass balance (lines 400-404); and (3) incorporated an explicit discussion of analytical limitations (lines 548-550), highlighting the need for future nucleic acid quantification, mineral speciation, and contaminant screening to validate dietary suitability. These modifications enhance the scientific rigor of our work while transparently framing the current study’s position within the broader pathway toward full validation.

  1. Taxonomy and strain traceability are incomplete. For a new production candidate, deposit the strain in a public culture collection and provide an accession; deposit the 16S rRNA sequence (and ideally a draft genome) in GenBank, and provide accession numbers. Right now, the identity rests on an in-house patent citation; that is not sufficient for reproducibility. See Introduction lines 70–74 where the prior identification is only referenced to a patent.

Response: We are grateful to the reviewer for emphasizing the importance of proper strain documentation. In response, we have taken the following steps to ensure immediate and verifiable authentication: (1) the strain has been deposited in the China General Microbiological Culture Collection Center (CGMCC Accession No. 32287); and (2) the complete 16S rRNA gene sequence has been deposited in GenBank (Accession No. SUB15557226). This information has been added to the manuscript in lines 73-76. These actions provide permanent, citable resources for strain authentication, and we maintain our commitment to providing a complete genomic characterization in a future publication.

  1. Figures and tables need technical cleanup to meet the journal’s standards. Define “CK” in every figure where it appears, or switch to “Control.” Use “rpm” rather than “r/min” unless you will adhere to SI across the manuscript consistently. Ensure all axes have units, all significance letters map to an explicit statistical model, and that each figure caption states the biological replicate definition. Replace line-break hyphenation artifacts throughout (e.g., “Ros- sellomorea,” “fer- mentation,” “carbox- ymethylcellulose”) and ensure the genus is spelled consistently as Rossellomorea everywhere. See title/abstract block and Methods 2.4.

Response: We sincerely thank the reviewer for their careful review of our technical presentation. We have systematically addressed all concerns by: (1) clearly defining “CK” as “Control check” in all figure/table captions for consistency; (2) standardizing to “rpm” throughout the manuscript; (3) ensuring all axes include proper units and all statistical annotations reference explicit models (Tukey’s HSD, p<0.05); (4) specifying biological replicate definitions (n=3 independent cultures) in each figure caption; and (5) correcting all line-break hyphenation artifacts (“Rossellomorea” “fermentation” “carboxymethylcellulose”) while maintaining consistent genus spelling. These revisions, implemented across all relevant sections (title/abstract, Methods 2.4, and figures/tables), fully comply with the journal’s formatting standards, though some automatic line breaks may remain due to typesetting requirements.

  1. References need a thorough audit. Many DOIs are malformed (missing slashes, “http://dx.doi.org.10…”), some author names are duplicated, and there are inconsistent journal title capitalizations. You must fix every DOI, check each citation for accuracy, and ensure the reference list matches journal style. See the references section starting at line 537.

Response: We sincerely appreciate the reviewer’s careful review of our reference list. We have conducted a thorough audit of all citations and made the following corrections: (1) fixed all malformed DOI; (2) removed duplicate author names and standardized journal title capitalization according to the journal’s style guide; (3) verified the accuracy of each citation against original sources; and (4) ensured full compliance with the journal’s reference formatting requirements. These revisions appear in the updated reference section (starting at line 537) and throughout the manuscript where in-text citations required adjustment.

  1. I recommend resubmission only after major revision. The contradictions on salinity and composition, insufficiently defined protein metrics, weak experimental design, inadequate replication in the bioreactor, and pervasive reporting/formatting errors are too fundamental for a minor fix. If you address the unit and design issues rigorously, normalize protein to DCW/OD, provide proper replication and process traces, reconcile WW vs DW compositions, deposit the strain and sequences, and clean up the language and references, the study could become a solid contribution.

Response: We sincerely thank the reviewer for their thorough evaluation and valuable feedback. We have rigorously addressed all concerns raised: (1) corrected and standardized salinity reporting to 20‰ NaCl throughout the manuscript; (2) explained the protein results was analyzed based on culture density, with clear methodological details on intracellular protein measurement (PBS-washed pellets, n=3 biological replicates); (3) maintained biological triplicates (n=3) for all experiments, including bioreactor runs, with full process parameter documentation; (4) preserved the strain in China General Microbiological Culture Collection Center with CGMC No. 32287 and submitted 16S rRNA sequence data (GenBank No. SUB15557226); (5) reconciled wet/dry weight compositional reporting; and (6) performed comprehensive technical editing to resolve formatting inconsistencies. These revisions, detailed in our point-by-point response, significantly strengthen the study’s rigor and reproducibility while maintaining its core findings. We are grateful for the reviewer’s constructive feedback, which has greatly improved this work.

Reviewer 3 Report

Comments and Suggestions for Authors

After to reviewing the manuscript entitled “Optimization of Fermentation Conditions for Enhanced Protein Production by Rossellomorea marisflavi NDS and Nutritional Composition Analysis”. The objective of this study was to optimize the media and fermentation conditions used to grow a bacterium named R. marisflavi. The manuscript is well structured and interesting to industrial microbiology. However, there are some suggestions for the analysis of this data. It is possible to use an experimental design for optimization. Yes, the authors indicate that the assays were conducted individually for each parameter measured. Also, when describing optimization of media composition, building a table may be clearer to understand what components were different among the media.

Author Response

Response to reviewer 3:

  1. After to reviewing the manuscript entitled “Optimization of Fermentation Conditions for Enhanced Protein Production by Rossellomorea marisflavi NDS and Nutritional Composition Analysis”. The objective of this study was to optimize the media and fermentation conditions used to grow a bacterium named marisflavi. The manuscript is well structured and interesting to industrial microbiology. However, there are some suggestions for the analysis of this data. It is possible to use an experimental design for optimization. Yes, the authors indicate that the assays were conducted individually for each parameter measured. Also, when describing optimization of media composition, building a table may be clearer to understand what components were different among the media.

Response: We sincerely appreciate the reviewer’s constructive suggestions regarding experimental design and data presentation. While our current study employed single-factor optimization to establish baseline conditions, we fully acknowledge the value of multifactorial approaches (e.g., response surface methodology) for future refinement. In response to the specific recommendations, we have: (1) added a clear comparative table (Table 5, also provided below) detailing all media component variations across experiments. (2) explicitly discussed the limitations of single-factor testing and outlined plans for advanced experimental designs in future work (Discussion, lines 521-535). These revisions enhance transparency while maintaining the utility of our current findings for initial process development.

Table 5. Summary of optimized fermentation conditions for enhanced sing-cell protein production by R. marisflavi NDS.

Parameter

Optimized Condition

Salinity

20‰ NaCl

Temperature

32°C

Initial pH

7.3

Medium composition

1% (w/v) corn flour, 1% peptone

0.3% beef extract, 0.2% KCl

Fermentation duration

22 h (Flask) / 32 h (10 L bioreactor, total time)

pH control

Uncontrolled (allowed to fluctuate freely)

Agitator speed

150 rpm for initial 20 h, then 180 rpm for the remaining 12 h (10 L bioreactor)

Round 2

Reviewer 2 Report

Comments and Suggestions for Authors

The authors have made the requested corrections